# A Data-Driven Fine-Management and Control Method of Gas-Extraction Boreholes

Xiaoyang Cheng [1,2,3] and Haitao Sun [1,2,3,*]

1.  China Coal Research Institute, Beijing 100013, China
2.  State Key Laboratory of the Gas Disaster Detecting, Preventing and Emergency Controlling, Chongqing 400037, China
3.  China Coal Technology and Engineering Group, Chongqing Research Institute, Chongqing 400037, China
*   Correspondence: dreamsht@163.com; Tel.: +86-02365239611

**Abstract:** In order to improve the efficiency of gas extraction in coal mines, a data-driven fine-management and control method for gas extraction is proposed. Firstly, the accurate prediction of coal seam thickness and gas content was used to evaluate the gas reserves. Based on the time relationship between mining activities and gas extraction, the calculation model of borehole distance in different extraction units is established, and the differential borehole design is realized. Then, a drilling video-surveillance system and drilling trajectory measurement device are used to control the drilling process and the construction effect. Finally, the model of extraction data-correction and the identification of failed boreholes is established, then the failed boreholes are repaired. The technology method provided in the paper has realized the fine control of gas-extraction borehole design, construction, measurement, and repair, and formed a more scientific gas-extraction borehole control technology system, which provides new thought for efficient gas extraction.

**Keywords:** gas extraction; borehole design; borehole repair; refined management





## 1. Introduction

As the largest developing country in the world, China has a rapidly growing economic aggregate and energy consumption. As the primary driving force of China's economic development, coal occupies an important position in the energy structure [1–4]. In order to realize the double effect of "resource + safety" in coal mining, gas drainage has become an effective means [5,6]. Although gas extraction has played a positive role in reducing gas accidents, the low permeability in China restricts the improvement of gas-extraction efficiency. Therefore, gas accidents still cannot be fundamentally contained [7,8].

In order to effectively improve the efficiency of coal-seam gas drainage, experts and scholars carried out in-depth research on the design, construction, and sealing of extraction boreholes. Li Yun [9] optimized plane borehole into stereo borehole by changing borehole layout and borehole angle, which increased the pure amount of gas drainage and reduced the danger of outbursts. By conducting a simulation study on the pressure relief gas drainage parameters by using Fluent software, Ding Yang et al. [10] analyzed the influence of single and interactive factors on the drainage effect, fitted the regression model between the drainage parameters of each layer and the gas concentration in the upper corner, and obtained the optimal drilling parameters of each layer, which provided a guiding basis for accurate and efficient gas extraction. Qin Wei et al. [11] studied the law of strata movement in the working face and the characteristics of pressure relief gas drainage in the adjacent layer, based on which he put forward six kinds of drilling hole layout schemes, and determined the optimum drilling parameters by Fluent software simulation. Li Hong et al. [12] put forward the integrated technology of drilling construction, protection, and sealing, which can effectively solve problems such as difficult drilling and poor gas drainage effect in soft fault outburst coal seams.

The sealing effect of boreholes is also an important factor affecting the efficiency of gas drainage, and the rapid decline of gas drainage concentration is mainly caused by both in-hole gas leakage and hole-side fracture gas leakage [13,14]. For this reason, HU Shengyong et al. [15], Zhang Yongjiang et al. [16], Wang Zhiming et al. [17], and Wang Hao [18] analyzed the gas-leakage mechanism of gas boreholes from different angles, and put forward the corresponding borehole sealing technologies, which are of guiding significance for the improvement of gas-drainage efficiency. Fu Jianhua et al. [19] proposed a bag-type grouting sealing method characterized by a bag-type grouting sealing device. Engineering test results showed that the effect of the bag-grouting sealing method is preferable to that of the traditional "two sealing and one grouting" method. Zhang et al. [20] proposed a new in situ failure control technology, using fine particles to seal the leakage cracks around the CMB to improve the gas drainage effect. Zhai Cheng et al. [21–23] developed composite sealing material, solidified sealing material, and flexible gel (FG), and Li Bo et al. [24] fabricated a new coal-dust polymer composite material (CP), all of which are of great significance for improving the sealing quality of the borehole. Fu Jianhua. et al. [25] obtained the optimal ratio of cement-based sealing materials by orthogonal experimental method, which improved the extraction rate, the configuration, and related characteristics of sealing materials. Zhang Tianjun et al. [26] focused on the effect of the proportion of expansion agent on the expansion and creep properties of cement-based sealing materials. In addition, Zhang Chao et al. [27], Ni Guanhua et al. [28], and Zhou Aitao et al. [29] also developed different kinds of sealing materials, which not only provide more material options for borehole sealing, but also effectively improve the borehole sealing quality, thus laying a foundation for efficient gas extraction. Zhao D [30], Zhang Kai [31], Li Pu [32], and Wang Kai [33], respectively, studied the reasonable sealing depth of gas-drainage boreholes under different geological conditions and different sealing techniques by means of theoretical analysis, numerical simulation, and industrial test, which improved the sealing quality of gas drainage boreholes. Cheng Zhiheng et al. [34] established a multi-factor evaluation of the hole sealing method, which provided a theoretical basis for the selection of hole sealing method. Niu Yue et al. [35] developed a set of test systems for monitoring borehole deformation during gas drainage, proposed the calculation index of borehole damage degree (DD) based on residual area, established the coupling relationship between relative pressure (RP) of sensor and DD, and modified the DD calculation model to determine the condition of borehole damage. The research results have certain guiding significance for the deformation measurement and damage judgment of gas-drainage boreholes.

Although experts and scholars have made positive contributions to the improvement of gas-extraction efficiency in many aspects, gas accidents in recent years have exposed new problems, such as construction deviation of gas-extraction boreholes, insufficient number of boreholes, and so on. For this reason, on the one hand, new technology and new equipment are adopted to effectively supervise the whole process of gas drilling construction. Zhao Enbiao et al. [36] used drilling peeping and trajectory measurement technology to observe the gas-extraction drilling hole, and determined the rationality of the drilling location. Zhang Jun [37] adopted the techniques of drilling trajectory measurement, real-time data transmission and three-dimensional visualization of data mapping to realize the visual supervision of drilling construction, which improved data transmission and processing efficiency. In order to improve the accuracy of borehole trajectory measurement, Yang Yi [38], Yang Hai [39], Liu Xiushan [40] et al. used different methods to correct the drilling trajectory, which significantly improved the accuracy of the borehole trajectory measurement. Zheng Lei [41] used a drilling video-surveillance system to realize parameter recording and video real-time monitoring in drilling process. In order to improve the clarity of the monitoring picture, Xu Yonggang et al. [42] proposed an adaptive compression and hybrid multiple hypothesis-based residual reconstruction algorithm based on a normalized Bhattacharyya coefficient (NBCAC-MHRR) to solve the high-efficiency video coding (HEVC) problem in underground coal mines, which improved the quality of underground video surveil-

lance in coal mines. On the other hand, the concept of fine management is applied to gas drainage management [43,44]. Through the establishment of fine management systems and assessment mechanisms, each link of gas drainage is effectively managed and controlled, so as to improve the supervision level of mine gas drainage and reduce the risk coefficient of gas drainage.

The above research improves the efficiency of gas extraction from many aspects. However, it is worth noting that coal mine gas extraction is a systematic project, and it is difficult to achieve high-efficiency gas extraction only by the improvement of a single technology. In this paper, the accurate assessment of gas reserves and the differential design of drilling holes are carried out in the early stage of gas extraction, and the effective control is carried out in the drilling construction process. In the process of gas extraction, data correction is carried out, and the failed drilling holes are identified and repaired, forming a gas-extraction-control technology system, combining technology and management, which provides a solution for efficient gas extraction.

## 2. Accurate Evaluation of Gas Reserves in Coal Seams

The evaluation of coal seam gas reserves is an important foundation for the design of extraction boreholes, and it is calculated according to the following formula:

$$G = 0.01AhDC \tag{1}$$

In the formula, $G$ is gas reserves of extraction unit, $m^3$; $A$ is area of extraction unit, $m^2$; $h$ is net thickness of coal seam of extraction unit, m; $D$ is coal seam bulk density of extraction unit, $t/m^3$; $C$ is gas content of extraction unit, $m^3/t$.

Formula (1) shows that the accuracy of gas reserves calculation largely depends on two parameters: coal reserves and gas content in the coal seam. However, in the actual calculation process, there is often a large deviation in the calculation of coal seam gas reserves due to the influence of coal seam occurrence state, geological structure, mining stress and so on, which seriously restricts the reasonable design of gas-drainage boreholes.

For this reason, a gas geological dynamic analysis system is developed based on the SuperMap platform. First of all, the system vectorizes the spatial information of the coal mine and forms the digital map of gas geology on the basis of the mining engineering plan map, topography map, and so on. Secondly, the gas geology-related information in the geological prospecting stage and production stage is uniformly stored and informationized, which lays the foundation for the deep utilization of the data. Finally, based on the organic correlation of coal mine spatial information and data, the automatic generation and dynamic updating of coal seam occurrence parameters are realized. The system makes full use of coal mine drilling, geophysical exploration, survey, and other multi-source data for fusion analysis, which reduces the analysis error caused by data singleness, data dispersion, and data missing, and improves the accuracy of gas geological information prediction. The intelligent extraction component of coal seam thickness and the intelligent update component of gas content provide technical support for the accurate evaluation of coal reserves and gas content.

### 2.1. Accurate Evaluation of Coal Reserves

The commonly used calculation methods of coal reserves include profile calculation method, geological block calculation method, horizontal section calculation method, statistical calculation method etc. [45]. These methods are all based on Formula (2):

$$Q = S \times M \times \rho \tag{2}$$

In the formula, $Q$ is coal reserves, t; $S$ is the scope of resources, $m^2$; $M$ is the thickness of coal seam, m; $\rho$ is the bulk density of coal $t/m^3$.

In the actual calculation, $S$ and $\rho$ are fixed values, so the accuracy of reserves calculation mainly depends on the accuracy of coal seam thickness M. However, affected by the

geological environment, tectonic movement, and other objective factors in the process of coal formation, there are significant differences in coal seam thickness [46]. In the gas geological dynamic analysis system, the inverse distance interpolation method as shown in Formula (3), is used to realize the accurate evaluation of coal seam thickness. On the one hand, based on the geological prospective borehole data, the intelligent generation of original coal seam thickness grid and isoline is realized. On the other hand, based on the measured coal seam thickness revealed by coal mining activities, the intelligent correction and update of coal seam thickness isoline are realized. The two work together to realize the accurate evaluation of coal seam thickness. The thickness grid and isoline of coal seam are shown in Figure 1.

$$Z*(x_0) = \frac{\sum\limits_{i=1}^{n} \frac{1}{(D_i)^p} Z(x_i)}{\sum\limits_{i=1}^{n} \frac{1}{(D_i)^p}} \tag{3}$$

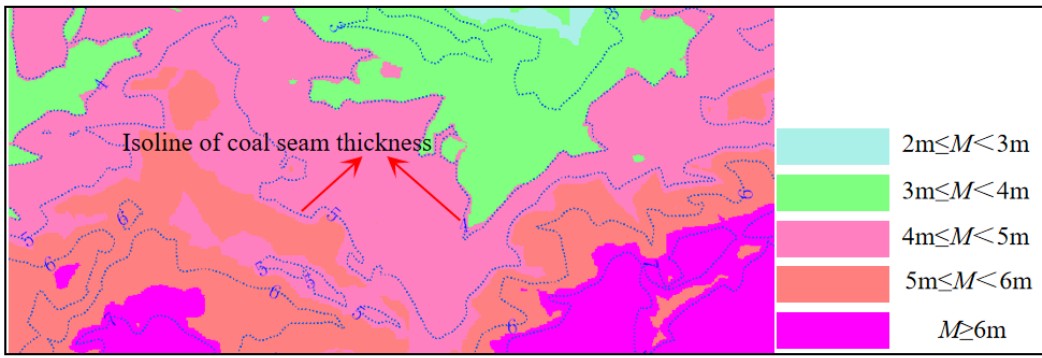

**Figure 1.** Coal seam thickness grid and isoline map of coal seam thickness.

In the formula, $Z*(x_0)$ is the calculated coal seam thickness of the interpolated point, $Z(x_i)$ is the coal seam thickness value of the $i$th ($i = 1, 2, \ldots \ldots, n$) known sampling point, $n$ is the number of sampling points used for coal seam thickness interpolation, $D_i$ is the distance from the interpolated point to the $i$th sampling point, $p$ is the power of the distance.

In order to further improve the calculation accuracy, the extraction unit can be divided into n blocks according to the differences of geological blocks, coal seam thickness and mining environment, and the coal reserves can be calculated in sequence. The calculation formula is shown in Formula (4).

$$Q = \rho \cdot \sum_{i=1}^{n} \iint\limits_{s_i} h \cdot d\sigma \tag{4}$$

In the formula, $S_i$ is the resource range of block $i$, m$^2$, $h$ is the thickness of coal seam, m.

### 2.2. Dynamic Evaluation of Gas Content

Gas content is also an important parameter for the calculation of coal seam gas reserves. In engineering practice, the prediction of gas content is often used as the calculation parameter of gas reserves.

However, in the process of coal seam mining, affected by the change of mining stress, the gas content of coal seam is in a constant state of dynamic change, so a large deviation is inevitable when using the original gas content to evaluate gas reserves. Based on the original gas content data, the gas geological dynamic analysis system generates the original gas content isoline, and uses the measured gas content measuring point data to calculate the Kriging interpolation method using the Formula (5). The prediction of gas content isoline

is intelligently updated, which greatly improves the accuracy of gas content prediction. The gas content isoline before and after update is shown in Figure 2.

$$\hat{Z}(S_0) = \sum_{i=1}^{N} \lambda_i Z(S_i) \tag{5}$$

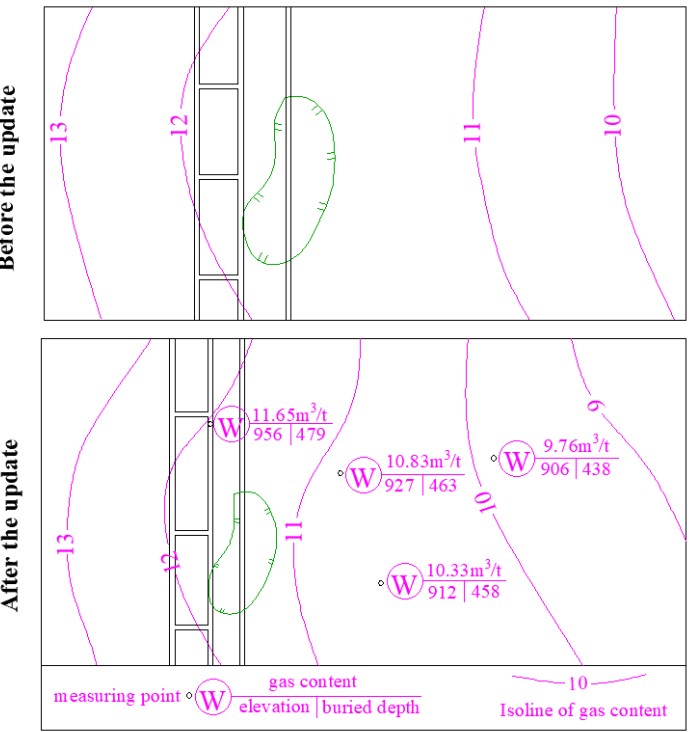

**Figure 2.** Comparison of gas content contour before and after update.

In the formula, the gas content value of the $i$th measuring point, the unknown weight of the $i$th content value, $S_0$ is the predicted position, and $N$ is the number of measuring points.

## 3. Differential Extraction Borehole Design

A reasonable drilling design of gas drainage needs to consider the position and time relationship between mining activities and gas drainage [47]. In the process of actual extraction borehole design, for the convenience of design and construction, unified drilling construction parameters are often used in the same working face, but the blind unity of borehole spacing can bring great safety risks and engineering waste to the mine. Therefore, for the rationality of extraction balance, this paper makes a differential design of extraction boreholes, that is, the design of sub-sections and non-uniform boreholes is adopted in the borehole design, and the schematic diagram of differential borehole design is shown in Figure 3.

The calculation formula of the amount of gas extraction required to reach the standard of gas extraction in the working face section is as follows:

$$Q_i = (W_{i\max} - W_0) \cdot \rho \iint\limits_{S_i} h \cdot ds \tag{6}$$

In the formula, $Q_i$ is the extraction amount per unit borehole length, m$^3$; $W_{i\max}$ is the maximum gas content in the $i$th section, m$^3$/t; $W_0$ is the critical gas content of extraction standard, m$^3$/t; $\rho$ is the density of coal, t/m$^3$; $h$ is the thickness of coal, m; $S_i$ is the cross-sectional area of the $i$th section, m$^2$; and ds is the section microelement.

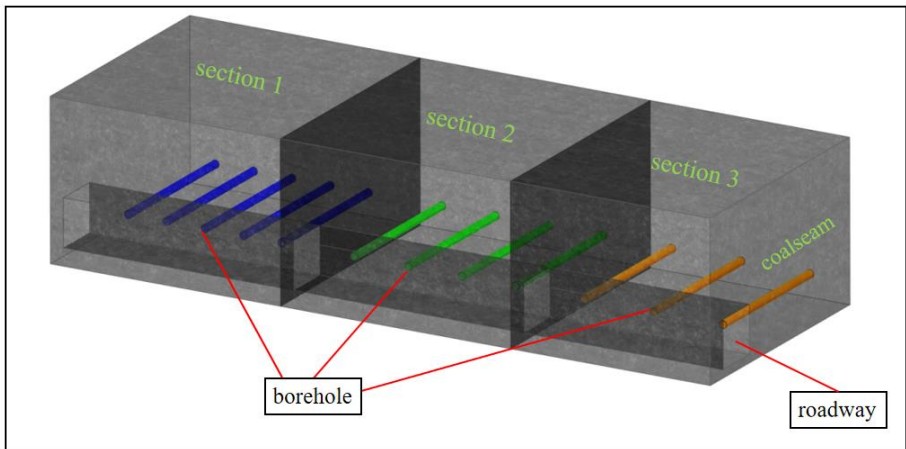

**Figure 3.** Schematic diagram of differential extraction borehole design.

The amount of gas extraction required to reach the standard in the $i$th section can be calculated by the amount of extraction per unit length of boreholes and the total length of extraction boreholes, which can be expressed as follows [48]:

$$Q_i = q \cdot L_i = \frac{Q_0(1 - Be^{-Ct_i})}{L_0} \cdot L_i \tag{7}$$

In the formula, $q$ is the extraction amount per unit borehole length, $m^3/m$; $L_i$ is the total length of extraction borehole in the $i$th section; $L_0$ is the length of single borehole, m; $Q_0$ is the initial extraction amount of single borehole, $m^3$; $B$ and $C$ are constants; the shortest extraction time of the $i$th section is $t_i$, and its calculation formula is as shown in the Formula (8):

$$t_i = \sum_{i=1}^{n} \frac{M_{i-1}}{Y} \tag{8}$$

In the formula, $M_{i-1}$ is the coal reserve of the $i-1$th section, t; $Y$ is the daily output of the working face, t/d.

Based on Formulas (6) and (7), the total length of extraction boreholes in $i$th section can be obtained:

$$L_i = \frac{(W_{i\max} - W_0) \cdot L_0 \cdot \rho \iint\limits_{S_i} h \cdot ds}{Q_0(1 - Be^{-Ct_i})} \tag{9}$$

The volume of coal in the extraction area can be expressed in two ways, which are approximately equal:

$$\iint\limits_{S_i} h \cdot ds \approx \pi R_i^2 \cdot L_i \tag{10}$$

In the formula, $R_i$ is the extraction radius, m.

Based on the Formulas (8)–(10), it is concluded that the extraction radius of the $i$th section is:

$$R_i = \sqrt{\frac{Q_0\left(1 - Be^{-C\sum_{i=1}^{n} \frac{M_{i-1}}{Y}}\right)}{(W_{i\max} - W_0) \cdot \pi \cdot L_0 \cdot \rho}} \tag{11}$$

In order to avoid production safety accidents caused by the extraction blanking zone, the effective extraction range of extraction boreholes should completely cover the extraction section. According to the Formula (11), the distance between gas drainage boreholes in the $i$th section can be determined as $2R_i$, which increases the rationality of gas drainage boreholes in the section with high gas and low drainage time, and reduces the engineering

waste in the section with low gas and high drainage time, thereby providing guidance for efficient gas extraction.

## 4. Management of Drilling Process and Drilling Effect

The efficient extraction of gas is closely related to the quality of extraction boreholes. In addition to the reasonable design of boreholes, precision borehole construction is essential. At present, there is a serious disconnection between borehole construction and borehole design, and drilling site constructors have the tendency of blindly pursue drilling footage due to the lack of supervision of drilling site construction, resulting in a discrepancy between drilling construction parameters and design, inadequate borehole construction and unclear description of the site and other major hidden dangers that threaten coal mine production safety. In order to improve the quality of drilling construction, it is necessary to effectively manage the drilling process and drilling effect.

(1)    Management of drilling process

In order to ensure the precise construction of drilling, it is necessary to timely acquire the drilling hole angle, hole position, drilling distance, and so on, to avoid safety accidents caused by deviation of drilling construction. As an effective method to manage drilling process, drilling video-surveillance systems are widely used in coal mines. The working principle of the system is to collect video information during the drilling process through an underground high-definition camera, and transmit the video signal to the ground server by optical cable transmission technology to realize the real-time monitoring of the drilling site. The network topology and monitoring effect are shown in Figure 4.

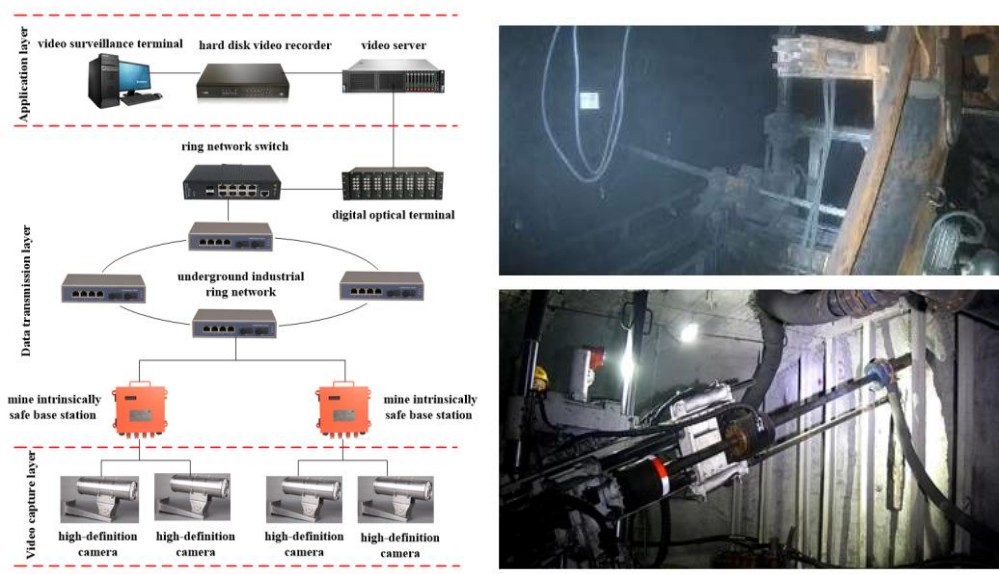

**Figure 4.** Network topology diagram of drilling video surveillance system and monitoring effect.

(2)    Management of drilling effect

In order to reduce the influence of gas-extraction borehole construction deviation on gas extraction, the drilling trajectory measurement device is used to measure the construction trajectory of gas-extraction borehole. In the process of drilling construction, the probe tube is installed in the non-magnetic drill pipe behind the drill bit, and the probe tube is drilled forward with the drilling rig. The basic data such as drilling depth, dip angle, and azimuth angle are measured and recorded by hand-held data storage instrument, and the data processing and automatic generation of drilling trajectory are carried out by professional software. The drilling trajectory measurement device and measurement results are shown in Figure 5.

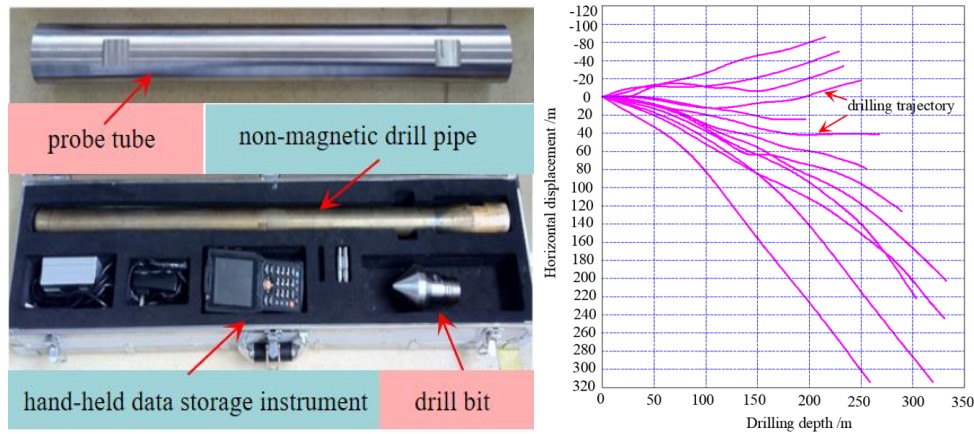

**Figure 5.** Drilling trajectory measurement device and measurement results.

## 5. Identification and Repair of Failed Gas Extraction Boreholes

### 5.1. Extraction Data Correction

Extraction data are generally measured by extraction monitoring system on-line or measured manually. Although advanced extraction monitoring system and measuring instruments are adopted, there are still technical difficulties to maintain the accuracy of the system and instruments in the long process of extraction, which inevitably results in the deviation between the extraction data and the real data. For this reason, the difference between gas extraction monitoring data and manual data is investigated, the deviation law of gas-extraction monitoring data is analyzed, the correction model of gas extraction data is established, and the automatic calibration of gas-extraction monitoring data is achieved using computer technology to improve the accuracy of mine gas-extraction monitoring data. The gas-extraction data correction process is shown in Figure 6.

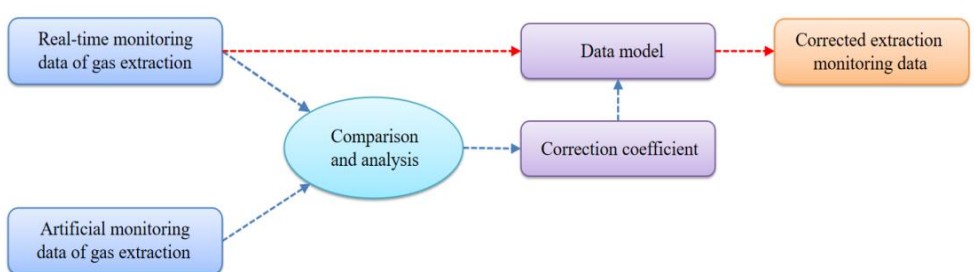

**Figure 6.** Flow chart of gas-extraction data correction.

In addition, based on the different characteristics of gas conditions in the gas pumping station and the extraction pipeline, the extraction gas quantity needs to be converted into the gas quantity in the standard state to be calculated. For the monitoring device which can directly display the gas quantity in the standard condition, the default parameter values (temperature, pressure, etc.) in the device should be updated in time according to the actual gas conditions of the equipment installation site. Finally, the accurate measurement of gas-extraction data is based on the accuracy of the sensor, and the sensor needs to be adjusted regularly and tested on the spot. If the test result deviates greatly, the sensor should be replaced or repaired in time.

### 5.2. Identification of Failed Gas-Extraction Boreholes

After the formation of gas-extraction boreholes, under the action of the stress field around the boreholes, the boreholes are often unstable, the cracks between the boreholes are connected, and the cracks between the boreholes and the surrounding rock are connected, resulting in the decrease or even interruption of gas drainage concentration and flow

rate [49,50]. Therefore, it is necessary to identify the failure gas-extraction boreholes so as to guide the restoration of the failed boreholes.

In the extraction pipe network, the gas confluence of each extraction hole enters the branch pipe of the extraction unit, and a flow sensor is installed on the branch pipe to measure the extraction flow of the whole extraction unit. The schematic diagram of the extraction system is shown in Figure 7. The pumping system only depends on the data on the branch pipe and the main pipe, which makes it difficult to grasp the extraction data and state of a single borehole in real time and accurately. Therefore, this paper adopts the data source processing method. When all the extraction boreholes of the extraction unit are connected into the initial stage of the extraction system, the gas-extraction parameter measuring instrument is used to measure the gas flow of each borehole. At the same time, the total flow of the extraction unit is read by the branch pipeline flowmeter, and the flow ratio of each drill hole is calculated by Formula (12).

$$\phi_k = \frac{Q_k}{Q_c} \tag{12}$$

In the formula, $\phi_k$ is the $k$ borehole flow ratio, $Q_k$ is the $k$ borehole flow, and $Q_c$ is the total branch flow.

In the initial stage of extraction, due to the pressure relief and permeability enhancement of boreholes and the negative pressure of extraction, the gas flow rate of each borehole is larger and the difference of flow ratio is small. With the extraction time lengthens, there are some unstable and air leakage boreholes one after another, and a large amount of air will be mixed in the gas extracted from these boreholes, which will be merged into the extraction pipeline to reduce the extraction flow [20]. At this point, according to the sub-source data processing method, the flow ratio of this kind of boreholes decreases significantly, while the flow ratio of normal borehole increases significantly. According to the extraction requirements of each mine, when the borehole flow ratio is lower than the critical value $\phi_L$, it can be judged as failed gas-extraction boreholes, and technical measures need to be taken to repair or close the extraction boreholes in order to achieve efficient gas extraction.

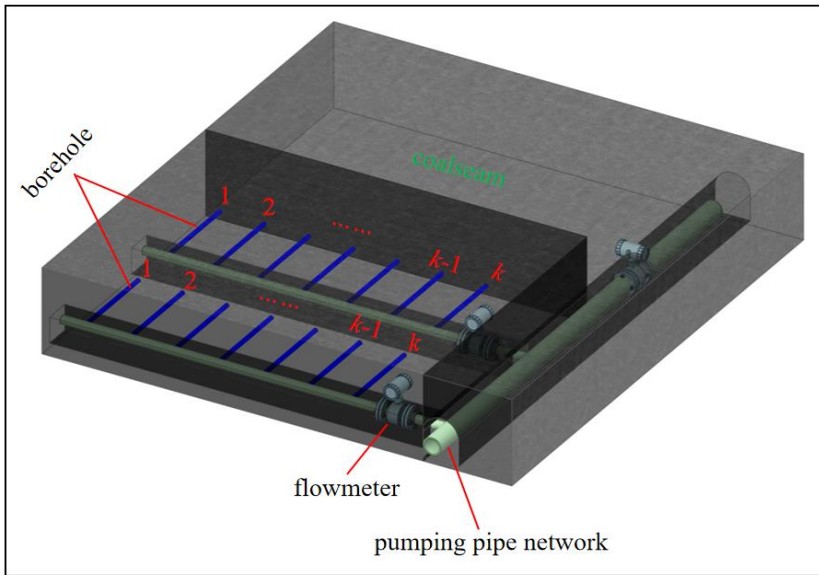

**Figure 7.** Schematic diagram of pumping pipe network.

### 5.3. Repair of Failed Gas-Extraction Boreholes

The study shows that the main reason for the failure of gas-extraction boreholes is air leakage in boreholes [13,14], while the air leakage in boreholes is caused by poor sealing quality. There are mainly two ways to deal with the failed gas-extraction boreholes: closing

boreholes and supplementing boreholes, which are passive ways to deal with the failed boreholes. Closing the failed boreholes increases the drainage time of regional coal seams, and extraction effect of supplementary boreholes is also poor. Therefore, in this paper, the repair technology for failure borehole is adopted to realize efficient gas extraction. The process of drilling repair is shown in Figure 8.

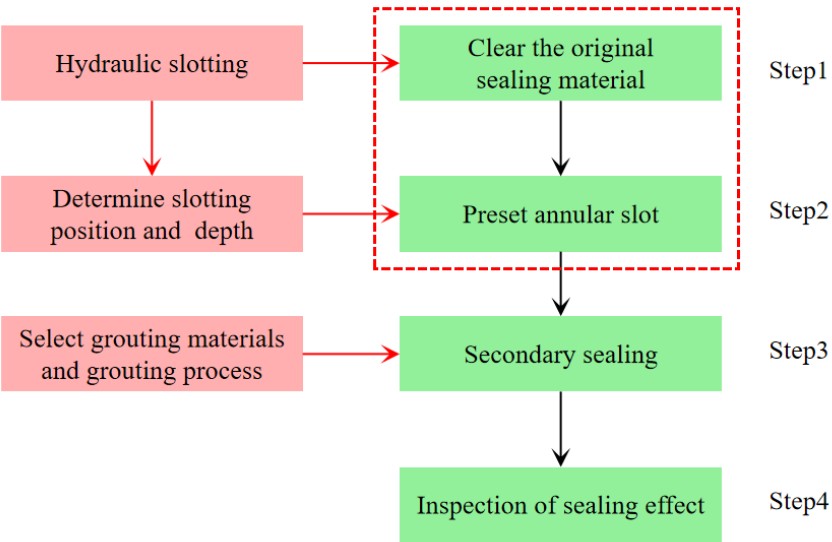

**Figure 8.** Repair process of failed gas-extraction boreholes.

As shown in Figure 8, firstly, the hydraulic slotting technology is used to cut and remove the failed sealing material in the borehole. Secondly, an annular slot is cut in front of the working face by a hydraulic slotting device to block the air leakage channel through the crack in the coal–rock mass. In this process, the optimal slotting position and depth need to be determined. Thirdly, secondary sealing is needed for the gas-extraction boreholes. Of course, in this process, the sealing material with high expansion and fluidity is essential, and it is recommended to adopt the "two plugging and one injection" sealing process with pressure. Finally, we need to check the sealing effect of the borehole.

## 6. Discussion

Gas drainage in coal mines is a complex process, including drilling design, drilling construction, extraction data management, and other links. With the wide application of information technology in the field of coal mines, professional software has been developed for different methods of gas extraction, which has played a positive role in improving the efficiency of gas extraction. Nowadays, in the context of the development of coal mine intelligence, the deep mining and utilization of data has become the core of coal mine intelligence development. As far as gas drainage is concerned, the intelligent control of gas drainage management based on extraction data, intelligent evaluation of extraction standards, and intelligent design of extraction boreholes have become an important but difficult part of the development of gas-drainage technology. Future studies should focus on building an intelligent decision-making platform for gas drainage based on gas-drainage data by means of big data analysis to achieve intelligent control, thereby realizing the intelligent construction of gas drainage.

## 7. Conclusions

(1) Through the accurate prediction of coal seam thickness and gas content, the accurate assessment of coal seam gas reserves is realized. Then, the extraction area is divided into sections, and based on the space–time relationship between mining activities and gas extraction, the calculation model of borehole distance in different sections is established, and the differentiated design of borehole is realized.

(2) The drilling video surveillance system and drilling trajectory measurement device are used to manage the drilling process and effect, respectively, which ensure the precise construction of boreholes.

(3) Based on the monitoring data of gas extraction, the model of extraction data correction and identification of failed borehole are established and the failed borehole caused by air leakage is solved by repair technology of hydraulic slotting and sealing, which improves the gas drainage efficiency.

**Author Contributions:** Conceptualization, X.C.; methodology, X.C. and H.S.; validation, X.C. and H.S.; formal analysis, X.C.; investigation, X.C. and H.S.; resources, X.C.; data curation, H.S.; writing—original draft preparation, X.C. and H.S.; writing—review and editing, X.C. and H.S.; visualization, X.C.; supervision, H.S.; project administration, H.S.; funding acquisition, X.C. and H.S. All authors have read and agreed to the published version of the manuscript.

**Funding:** This research was funded by National Natural Science Foundation of China (No. 51874348), Chongqing Science Fund for Distinguished Young Scholars (No.cstc2019jcyjjqX0019), Natural Science Foundation of Chongqing (CSTB2022NSCQ-MSX1080).

**Data Availability Statement:** All data and/or models used in the study appear in the submitted article.

**Conflicts of Interest:** The authors declare no conflict of interest.

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
