# Peer review of "A Data-Driven Fine-Management and Control Method of Gas-Extraction Boreholes"

_processes, doi:10.3390/pr10122709_

Round 1

Reviewer 1 Report

This paper proposes a refined control method for gas drainage drilling based on data drive,which be of great significance to improve the efficiency of gas extraction. However, there are still some issues that should to be modified:

1.      The writing language needs polishing, some sentences are too long to read and difficult to understand precisely what the authors are trying to say.

2.      The abstract isn't clear and need to be improved with consist

(1) background

(2) objective

(3)material &method

(4)result & discussion

(5)Conclusion.

3.      The legend of the figure 1 needs to be added.

4.      In section 4, the paper mainly describes the control method of the drilling construction process, rather than the precise construction process of drilling. It is suggested to modify the title of chapter 3.

5.      It is recommended to add references in section 5.3.

6.      In section 6, this paper only describes the functions of gas extraction management and control platform, but does not elaborate the significance of the platform application. It is suggested to add this part.

Reviewer 2 Report

Manuscript is not good enough to be published in this journal. The author introduced a data-driven fine management and control mode of gas extraction borehole, However, this article lacks specific scientific details, and there is no academic innovation in academic. The manuscript is more like a technical introduction than a scientific article.

1.Relevant formulas are given in the paper, but there is no case to apply these formulas to obtain corresponding results.

2.Too many references in Chinese.

3.This article mainly introduces relevant equipment and technology, and lacks relevant application cases and data.

4.The differential extraction borehole design should be described in detail, and the relevant parameter acquisition process should be given.

5.The meaning of the numbers in Figure 2 should be explained.

6.The figure of borehole trajectory should be given.

Reviewer 3 Report

This is the review manuscript entitled “A data-driven fine management and control mode of gas extraction borehole”. I read the article carefully. Although the topic is interesting, several issues in this version of the manuscript have been addressed as follows.

1. The abstract should be rewritten carefully. Please remove all general information and describe your results in detail.

2. The keywords in the paper need to be further refined.

3. Please check carefully whether part of the measuring point data is missing in Figure 2.

4. Formulas should be referenced in the text.

5. It is suggested that Part4.1 focus on describing the drilling track measurement process rather than measuring equipment.

6. It is of great significance to repair the failed borehole after identification for efficient gas extraction. It is suggested to add this part.

7. The conclusion of the paper needs to be modified according to the revised content.

Round 2

Reviewer 2 Report

Incorrect reference number.

Author Response

The reference number in the paper has been completely modified, and some careless mistakes have
also been modified. You can seen the detailed content in the revised manuscript.

Reviewer 3 Report

The author has answered all the questions raised by reviever.

Author Response

The reference number in the article has been completely modified, and some careless
mistakes have also been modified, you can see the detailed content in the revised
manuscript.